# Surgical referral systems in low- and middle-income countries: A review of the evidence

Chiara Pittalis[1]*, Ruairi Brugha[1], Jakub Gajewski[2]

**1** Department of Epidemiology and Public Health Medicine, Royal College of Surgeons in Ireland, Dublin, Ireland, **2** Institute of Global Surgery, Royal College of Surgeons in Ireland, Dublin, Ireland

* chiarapittalis@rcsi.ie

**Data Availability Statement:** All relevant data are within the manuscript and its Supporting Information files.

**Funding:** The author(s) received no specific funding for this work.

## Abstract

### Background

Referral networks are critical in the timely delivery of surgical care, particularly for populations residing in rural areas who have limited access to specialist services. However, in low- and middle-income countries (LMICs) referral networks are often undermined by systemic inefficiencies. If equitable access to essential surgical services is to be achieved, sound evidence is needed to ensure efficient patient care pathways. The aim of this scoping review was to investigate current knowledge regarding inter-hospital surgical referral systems in LMICs to identify the main obstacles to their functioning and to critically assess proposed solutions.

### Methods

MEDLINE, EMBASE and Global Health databases and grey literature were systematically searched to identify relevant studies. The search generated 2261 unique records, of which 14 studies were selected for inclusion in the review. The narrative synthesis of retrieved data is based on a conceptual framework developed though a thematic analysis approach.

### Results

Multiple shortages in surgical infrastructure, equipment and personnel, as well as gaps in surgical and decision-making skills of clinicians at sending hospitals, act as obstacles to safe and appropriate referrals. Comprehensive protocols for surgical referrals are lacking in most LMICs and established patient pathways, when in place, are not correctly followed. Interventions to improve coordination and communication between different level facilities may enhance efficiency of referral pathways. Strengthening capacity of referring hospitals to manage more surgical conditions locally could improve outcomes, decrease the need for referral and reduce the burden on tertiary facilities.

### Discussion

The field of surgical referrals is still an uncharted territory and the limited empirical evidence available is of low quality. Developing strategies for assessing functionality and

**Competing interests:** The authors have declared that no competing interests exist.

effectiveness of referral systems in surgery is essential to improve access, coverage and quality of services in resource-limited settings, as well as overall health systems performance.

## Introduction

The patient referral system has been defined as *'a mechanism to enable comprehensive management of clients' health needs through resources beyond those available where they* [initially] *access care'* [1]. It links primary care facilities, district hospitals, provincial/regional hospitals and national referral centres, thereby facilitating the forward and backward movement of patients (including flow of information and documentation) according to the type of clinical expertise and management required [2]. As such, the referral system is pivotal in maintaining efficiency in allocation and utilisation of resources across the different levels of the health system and in providing a continuum of care, appropriate to patients' needs [3].

In low- and middle-income countries (LMICs) demand for surgical care, particularly specialist care, greatly outstrips provision, due to the critical shortage of a skilled health workforce outside of the main urban centres, leaving rural populations underserved [4]. Referral networks are essential in addressing patients' surgical care needs, and to reduce the incidence of morbidity and mortality from treatable surgical conditions. However, referral networks in LMICs are often undermined by systemic inefficiencies such as lack of transport, infrastructure [4] and sufficient skills at rural district hospitals [5], and poor coordination between different level hospitals [6–8], among other factors. Another widespread problem is the bypassing of lower level facilities by patients with simple conditions who self-refer directly to higher level hospitals for convenience, fear of delay in accessing the care they need, and belief that quality of care and more effective interventions are available at higher levels [3,9]. As a result, referral hospitals in LMICs are often congested with high volumes of patients with low-complexity conditions, and resources and staff time are absorbed by cases that could have been handled at lower levels [3,7,10,11]. This negatively impacts on costs (for patients and the health care sector) [8,12], waiting times and clinical outcomes, where appropriate management is delayed [5,13,14].

Despite growing attention to hospital surgical capacity in LMICs [8], referral systems and the interface between the different level hospitals have been under-researched [4,11,15]. If equitable access to essential surgical services is to be achieved, sound evidence is needed to ensure efficient patient care pathways [4,16]. This review investigates current knowledge regarding surgical referral systems in LMICs in order to: i) identify the main obstacles to their functioning effectively; and ii) identify proposed and tested solutions, and the available evidence of their effectiveness in addressing these gaps. The focus is on inter-hospital referrals from district hospitals to higher levels as district hospitals are the frontline providers of non-specialist surgical care for rural populations [4]. As such, they should have the infrastructure, resources and skilled clinicians to perform essential surgery and to decide on appropriate referrals.

## Methods

The research question for this study was 'what is known about inter-hospital surgical referral systems in LMICs?' A scoping review was selected as the most suitable approach for the study

as our objective was to produce a broad overview of the field and available research evidence [17,18].

The methodological approach was informed by multiple sources. The study design follows the theoretical principles underpinning systematic reviews described by the Cochrane Collaboration [19] and Petticrew and Roberts [17]. The selection of included papers and their qualitative assessment was guided by review manuals and tools published by the Joanna Briggs Institute [18,20]. The reporting framework follows the Preferred Reporting Items for Systematic Reviews and Meta-Analyses (PRISMA) approach [21] and its checklist for scoping reviews [22] (provided in S1 Table).

## Selection criteria

The Population, Concept and Context (PCC) framework suggested by the Joanna Briggs Institute [18] was used to support the selection of studies to be included in this review. Inclusion/exclusion criteria are summarised in Table 1.

Surgical referrals in conflict settings are determined by specific circumstances, surgical skills and responses, which are often not generalisable to stable settings, and were considered outside the scope of this review. Similarly, surgical referrals for safe induced abortions are governed by a range of legislative, cultural and ethical norms which are highly dependent on local contexts; and such studies were also excluded.

Sources of evidence: all study designs were considered, but text and opinion evidence was excluded. The search was restricted to studies in English language and publication date after 1990.

## Search strategy

The search strategy was informed by an extensive literature scoping exercise, with advice from bibliographic specialists and health systems experts. Its final structure was built around three main arms, namely:

- Terms relating to hospital referrals

- Terms relating to surgery

- Terms relating to LMICs/developing economies (following the World Bank classification system [23]).

The preliminary scoping exercise revealed that many published papers on referral systems focus on individual disciplines of surgery. Hence the search approach was designed to retrieve

**Table 1. Population, concept and context framework.**

|  | Inclusion criteria | Exclusion criteria |
|---|---|---|
| Population | Patients and staff involved in inter-hospital surgical referrals | Non-surgical referrals, intra-hospital referrals, community referrals and self-referrals |
| Concept | The functionality of referral networks, including referral patterns, pathways and obstacles | Papers where surgical referrals are mentioned but the main focus is not on the functionality of the referral system (e.g. papers depicting the range of surgical procedures performed or examining clinical and epidemiological aspects of referrals) |
| Context | • District, secondary and tertiary hospitals<br>• LMICs | • Primary care non-surgical facilities (e.g. health centres and dispensaries)<br>• Non-LMICs |

variations on the term 'surgery', for example 'obstetric surgical procedures' and 'orthopaedic procedures'. To maximise sensitivity, we included both medical subject headings (MeSH terms) and text words, combined using the Boolean operator 'and'.

The search strategy was executed through the examination of the bibliographic databases MEDLINE (PubMed), EMBASE and Global Health on 19–21 July 2018. It was initially tested in the MEDLINE database, reviewed and then adapted for the other databases. The original search string is reported in S2 Table.

A learning experience from the search process worth noting is that our search string was perhaps too broad, given the number and type of papers initially retrieved. Considering this is a new area of research (at least in LMICs) and that our scoping exercise demonstrated that such studies tend to be categorised under surgical sub-categories, our approach was to purposely keep the search broad to avoid the risk of omitting important studies, but there might be room for further refinement in future reviews.

### Other sources

The bibliographic search was complemented by thorough grey literature retrieval mechanisms, following international best practices and guidelines [24]. The grey literature search involved examining the websites of relevant multilateral and specialist organisations (e.g. World Health Organization) as well as using the Google search engine, with the same keyword search strategy. A snowball technique was also applied to scan reference lists of papers and other material to identify additional relevant publications. A further 15 studies were identified through this process and included in our review.

### Document management

The results of the search outputs were managed using the reference manager software Mendeley, which also supported identification and removal of duplicates. In order to keep accurate records of the review process, retrieved papers were imported into Covidence, a web-based platform for systematic literature reviews facilitating screening, quality appraisal and analysis.

### Screening

Two researchers independently reviewed each title and abstract, with a third researcher consulted in case of conflicting opinions. An inclusive approach was utilised in this first screening phase to ensure papers were not prematurely excluded. Following agreement on all potentially relevant titles and abstracts, full-texts of available selected papers were retrieved for the second round of screening. Again, two researchers reviewed and discussed the suitability of each paper in line with the PCC criteria in Table 1, with the mediation of a third assessor when needed. If a paper was excluded, reasons for exclusion were recorded and categorised as follows:

- Wrong scope (focus of study not on examining the functionality of the referral process)

- Wrong setting (study on referral system at community level, intra-hospital or in a high-income country)

- No full text available

- Text and opinion evidence

Covidence contains pre-set forms to facilitate quality assessment, but the forms are more suited for clinical trials. Since the evidence for this review was of a different nature, the

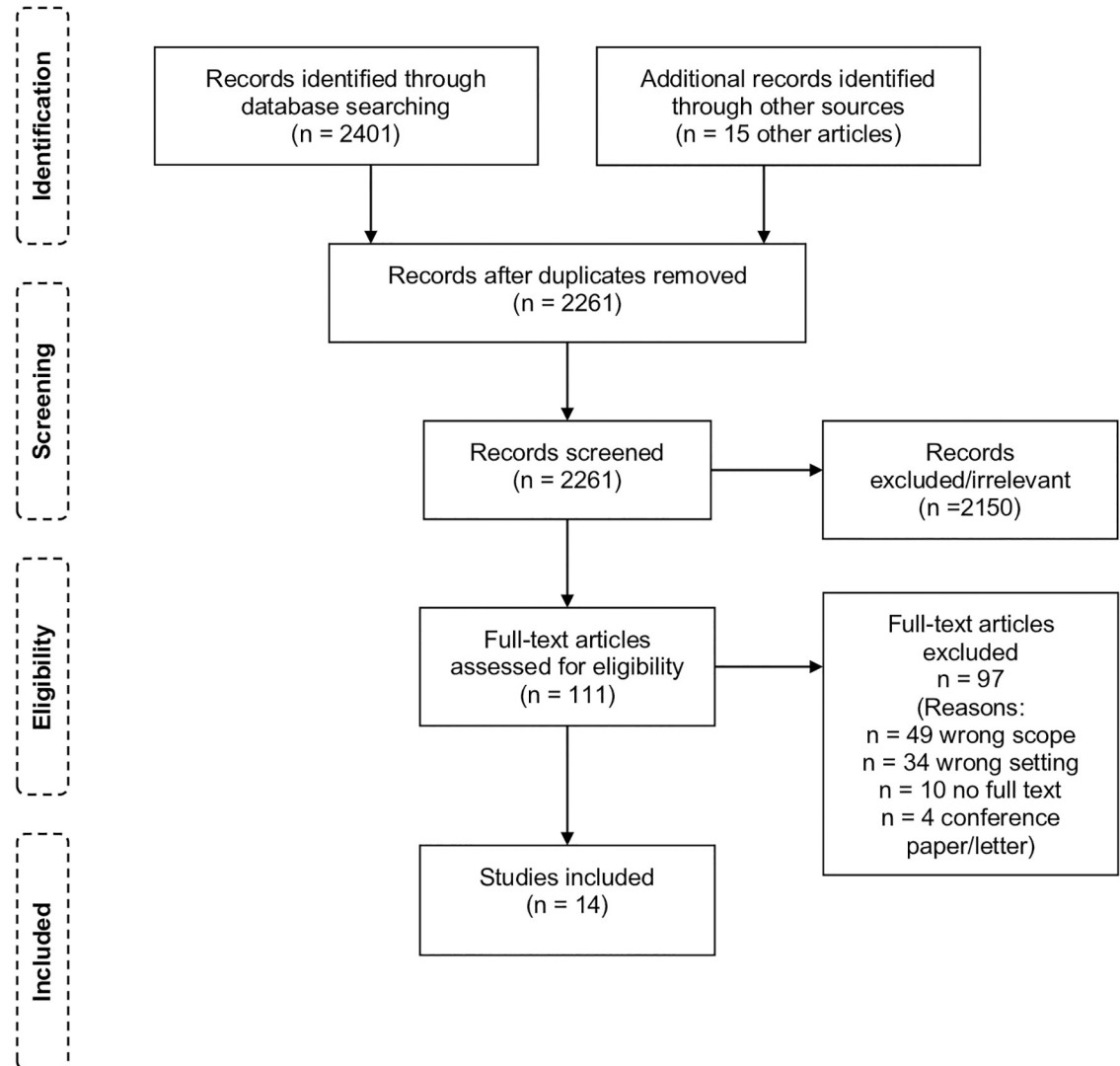

**Fig 1. Flow diagram of the screening process.**

assessment of each study was recorded separately, using The Joanna Briggs Institute Critical Appraisal tools [20]. Fig 1 shows a flow diagram of the screening process [21].

The initial search yielded 2261 records after duplicates were removed. Of these, 2150 were excluded by reading title and abstract, and 111 studies were considered for full text review. The final selection identified for inclusion in this review consists of 14 papers.

## Data charting and analysis

To our knowledge, there are no guidelines or validated tools for the assessment of inter-hospital surgical referral systems in LMICs. We therefore used a bottom-up thematic analysis approach [25] to identify emerging themes from the selected papers, thereby developing a conceptual (logic [26]) framework to guide our analysis and synthesis of findings.

We started by reviewing included papers to find any factors that might influence the referral process, loosely informed by Thaddeus and Maine's 'three delays model' of obstacles to the provision of adequate care in developing countries [27]. Firstly, we coded all factors mentioned

**Table 2. Emerging themes.**

| Emerging themes | Albutt et al 2018 | Crandon et al 2008 | Den Hollander et al 2014 | Goodman et al 2017 | Gyedu et al 2015 | Khan et al 2013 | Lee 2008 | Nkurunziza et al 2016 | Simba et al 2008 | Siraj et al 2016 | Fleming et al 2017 | Rudge et al 2011 | Sani et al 2009 | Shi et al 2014 |
|---|---|---|---|---|---|---|---|---|---|---|---|---|---|---|
| Availability of resources (supplies–incl. blood, equipment, staff) at sending facilities | x | | x | x | | x | | | x | x | | | x | |
| Skills level at sending facilities | x | | x | | | | | | x | x | x | | x | x |
| Referral by junior staff | | x | | | | | x | | | | | | | |
| Communication and coordination practices across facilities | x | | | | | | | | | | | | | x |
| Patient transport | x | x | | | | | | | | | | | | |
| Quality of referral documentation from sending hospitals | | x | | | x | | | | | | | | | |
| Patient management by sending hospital (incl. observance of safety measures pre- and during transfer) | | x | | | | | x | | | | | | | |
| Appropriateness of referrals (complexity of case, accuracy of diagnosis etc.) | | | x | x | | | x | | | | | | | |
| Timeliness of referral | | | x | x | | | x | x | | | | | | |
| National policies and guidelines | | x | | | | | | | | | | | | |
| National health priorities and funding | x | | | | | | | | | | | | | |
| Socio-cultural aspects of patient health seeking behaviour | x | | | | | | x | | | | | | | |
| Patients' financial resources | | | | | | | | x | | | | | | |

by the 14 papers as having a direct or indirect effect on referrals. Secondly, we combined conceptually-related ones under themes. Thirdly, we summarised citation frequency, as shown in Table 2. Emerging themes were then grouped into three broad categories: health facility factors, health system factors and operating environment factors.

Subsequently, we examined the available evidence to understand how identified parameters affect the functionality of the referral process. Type of outcome measures considered included referral patterns, timeliness, appropriateness and quality. Finally, we reviewed the studies to identify any objective measures of effect of the referral process on patient outcomes. The final analytical framework is illustrated in Fig 2 below.

Data charting forms were designed to extract relevant information on characteristics of reviewed papers (author, publication year, study country, population, setting, design and

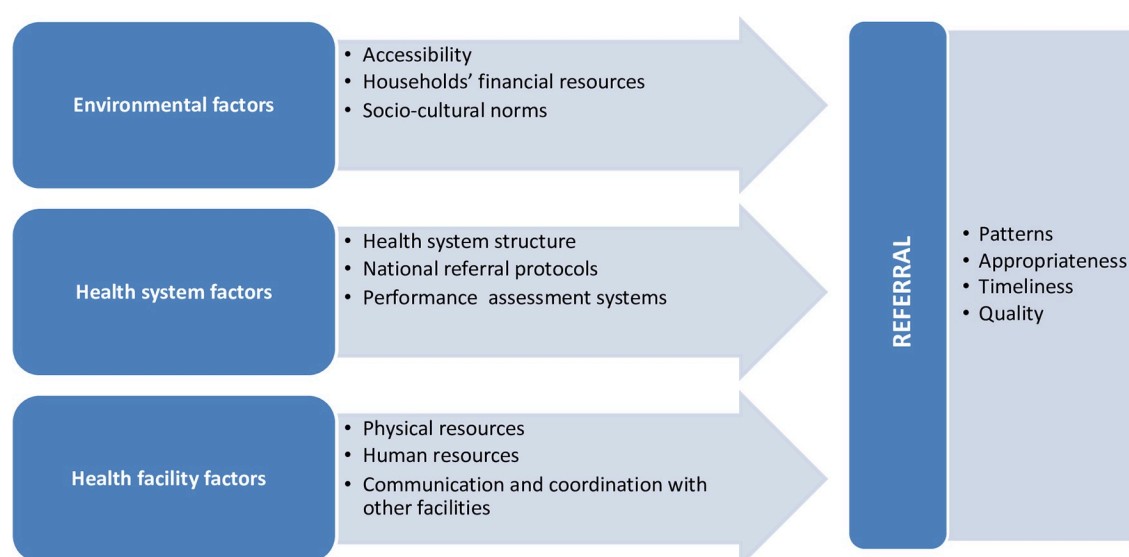

**Fig 2. Conceptual framework for the analysis.**

rigour) and outcome measures in a standardised format. Researchers filled in the forms in a collaborative manner, discussing results to agree on final content. The forms were updated in an iterative way, reflecting the analytical framework development process.

A note on terminology: the classification of district hospitals varies from country to country, they can be categorised as level one facilities (e.g. Tanzania) or level two facilities (e.g. Pakistan) depending on the structure of the national health system. However, the role of the district hospital as the first level facility at which basic general surgical procedures, such as C-sections and hernia repairs, can or ought to be performed is consistent across LMICs. In this manuscript we refer to district hospitals as first level surgical facilities, regional hospitals as secondary level surgical facilities and national referral hospitals as tertiary level surgical facilities in line with international conventions to describe the different levels of surgical care [4].

## Results

### Characteristics of included studies

Seven studies were conducted in Africa, five in Asia, and one each in Latin America and the Caribbean region. Publication dates ranged from 2008 to 2018. We identified potentially relevant studies conducted prior to 2000, but full texts were not available at the time of the review.

The study population for most papers consisted of patients, three investigated the perspectives of hospital staff. Six studies examined all surgical referrals, while others focused on specific areas of surgery, namely: obstetric (3), trauma (2), burns (1), paediatric (1) or cancer (1) surgery.

The majority of included studies (10) are descriptive in nature. Four papers report on the evaluation of interventions, mostly following a case study approach, with only one study involving a control group but in a non-randomised assessment. Based on international frameworks and standards for grading the strength of study design [28], the quality of evidence identified through our literature review is low. A summary of characteristics of the papers, including an overview of study design, rigour and JBI level of evidence [29], is provided in S3 Table.

The following sections present a narrative synthesis of retrieved data, as the nature and heterogeneity of the studies did not lend themselves to quantitative estimates of effects [17]. Results are reported according to the framework in Fig 2. Our analysis aimed to identify common themes across the studies, assessing similarities and differences in findings where possible.

## Factors affecting referrals

**Health facility factors.** At the level of individual health facilities, availability of essential resources, both physical (i.e. surgical and diagnostic equipment, supplies and infrastructure) and human (staff numbers and skills to manage cases), as well as communication and coordination with other facilities, were identified as the principal factors affecting referral patterns and practices.

Seven papers [5,7,11,13,14,30,31] in our review document lack of supplies, equipment and personnel as the main drivers of referrals to higher level facilities. In a study in the Dar es Salaam region of Tanzania over half of interviewed clinicians at sending health facilities reported referring patients, including surgical patients, due to lack of drugs (53.8%) and space (50%) [11]; while a study of incoming surgical emergencies at a tertiary hospital in Pakistan, recorded 85.7% of cases being referred due to lack of satisfaction with surgical services at local facilities [5]. Also in Pakistan, Siraj et al [30] report lack of blood products and intensive care units as key reasons for post-partum referrals (18 out of 23 patients); and non-availability of doctor, anaesthetist, blood bank and proper fetal monitoring equipment at district hospitals as the main reasons for intra-partum referrals. Similarly, service providers interviewed as part of a study in Uganda reported having to refer patients for lack of drapes, gowns, surgical blades and blood products [7]. In Tanzania over 96% of sending facilities reported referring for lack of expertise and equipment [11]. Resource shortages affect referrals at higher level facilities too. While the extent of shortages at secondary and tertiary hospitals is not quantified in the reviewed papers, there is some evidence of limited availability of very basic materials. For instance, in South Africa general surgical units referred patients due to lack of simple dressings for burns [14].

Surgical workforce weaknesses driving referrals comprise shortages in both staff numbers [5,30] and expertise [5,7,11,15,30,31]. Reported gaps include limited diagnostic capacity, ability to stabilise patients [15] and to handle certain surgical cases, such as trauma [31], emergencies [5] and high-risk patients [30].

In the reviewed literature there is also evidence of problems related to communication between sending and receiving hospitals. For instance, Albutt et al [7] described the referral system in Uganda as disorganised and uncoordinated, with generally poor communication during transfer of patients. Two papers [32,33] explored communication in more depth, by examining the quality of referral documentation from sending hospitals. A study of 643 elective surgical cases transferred to a tertiary hospital in Ghana found that none of the referral records included all required essential information, with 50% more chances of incomplete information when non-structured forms were used by referring clinicians [32]. A study of critically ill trauma patients in Jamaica reported major gaps in key patient information from referring facilities, such as time of injury (documented in 15.6% of cases), or monitoring of patients' clinical status (e.g. pulse rate in 13.1% of cases, Glasgow Coma Score in 19.7%) [33].

Three studies [7,30,33] captured issues with coordination and safe transport of patients across facilities. These ranged from lack of vehicles and fuel at the sending facilities [7,30], to lack of monitoring equipment in vehicles [30,33], and the transfer of critically injured patients

by drivers with no emergency medical technician training and without accompanying clinicians [33].

**Health system factors.** Our analysis considered evidence in relation to the health systems within which health facilities work. Health systems design and structure shape the organisation of health services at different care levels, including the patient care pathway, and the distribution of resources across them. While none of the reviewed studies looked specifically at this area, as mentioned by Albutt et al [7] surgery is not typically among the top priorities for local governments, with implications for funding and, in turn, hospital resources and capacity to deliver surgical services. For instance, at the time of their study Simba et al [11] reported that the Dar es Salaam region of Tanzania had only three district hospitals and no secondary hospitals. This contributed to overstretching available bed capacity at district level, driving patients to seek quite basic surgical care directly from the national hospital. Other studies mentioned lack of national guidelines or protocols to guide the referral process as important gaps [33].

**Environmental factors.** A number of studies pointed to the influence of wider country contexts on the functionality of referral systems, particularly in terms of patients' health seeking behaviour and utilisation of health services at the intended level of care. Lee [34] reported that parents' objection to transfer or further investigation affected referrals in 8% of the neonatal cases in his study. Albutt et al [7] documented patients bypassing lower level facilities, preferring to access care directly at higher level, while Siraj et al [30] found evidence of patients explicitly requesting to be referred from district hospitals to higher levels for social and personal reasons. Three studies [5,11,35] mentioned that financial constraints influence patients' health seeking choices and referral decisions.

## Implications for the referral process

The late referrals of patients, often when their clinical status has already deteriorated, is a common theme in the reviewed literature [5,7,13,14,34,35]. Den Hollander's study [14] on patterns of burn referrals in South Africa recorded one quarter of patients being referred one-week post-burn. In Malaysia, Lee [34] reported that prompt referral of infants with neonatal cholestasis was done in less than half of cases (47%), although early warning signs were common, including in six infants who had neonatal acute liver failure. In Nkurunziza [35] study of trauma in Rwanda, 47% of cases experienced delayed referrals.

Some of the papers investigated the reasons behind late referrals. In Lee's study [34] the delay was caused by the fact that health care providers who saw the cholestatic infants prior to referral failed to realise the seriousness of the condition. These included junior doctors at government district hospitals (in 6 cases), as well as senior doctors and paediatricians at private and government hospitals (12 cases). Investigation results not properly reviewed and acted upon, inconclusive biopsy results and incorrect diagnoses at the sending facilities also contributed to referral delays. Nkurunziza's study [35] offers good insights into the reasons for delayed referrals from the point of view of sending institutions, in this case district hospitals. For the 58 trauma referrals for which information on the reason for the delay was available, the main reported delay factors were patients awaiting appointment at receiving hospitals (44.8% of cases); lack of bed space at receiving hospitals (39.7%) and patients' financial challenges (13.8%). In Rwanda, even if patients had health insurance, they still had to provide a co-payment for ambulances and other medical expenses, on top of personal costs associated with the referral, contributing to delays or avoidance of referrals.

Two papers [13,14] documented inappropriate referrals. Den Hollander [14] reports that a large proportion of cases (44% of adults and 30% of children) referred by general surgical units to a specialised tertiary hospital could have been handled at their level of care. Goodman study

of obstetric referrals in Ghana, while not specifically assessing clinical appropriateness of referral, found potential indicators of unnecessary transfers, namely: 41 of 90 patients referred for prolonged labour had intact membranes; 25 of 90 parturients with "big baby" diagnosis had fundal height < 40 cm, and 13 of 139 with diagnosed hypertension had normal blood pressure on arrival [13]. The implication of such referrals was that operative interventions were needed.

Unnecessary referrals contribute to overwhelming referral hospitals [7,13,14], reducing their ability to cope with demand and provide specialist surgical care.

**Patient outcomes.** The types of clinicians making referrals at sending hospitals vary in cadres and seniority levels, but there is mixed evidence on patient outcomes. In Jamaica, Crandon et al [33] reported that most (93.5% of) referrals are made by junior clinicians, which may be a factor in the poor pre-referral patient stabilisation and management documented in their study. In contrast, in Malaysia Lee [34] observed that both junior and senior clinicians were responsible for misdiagnoses and delayed referrals.

Delayed referrals had negative consequences for surgical patients. Den Hollander [14] reported greatly reduced benefits and high risk of infection for burn patients, from referral to a specialist burn unit seven days after the occurrence of the burns. Crandon et al [33] suggested that skills gaps at sending facilities, as well as lack of referral protocols, may have contributed to the absence of basic stabilisation and safety measures pre- and during patient transport observed in their study (e.g. 55% of road traffic victims did not have cervical immobilisation during referral).

Two studies [5,34] quantified the effects of dysfunctional referral system on patient outcomes. Khan's study [5] of surgical emergencies at a tertiary referral hospital in Pakistan found that patients transferred from other health facilities had poorer vital statistics and experienced more deterioration of their clinical condition than those who arrived directly, with significant differences in length of stay in the intensive care unit (5.6 vs 1.5 days). However, there may be questions regarding case-mix comparability. Lee [34] study of infants with neonatal cholestasis found that late referral contributed significantly to adverse outcomes, especially in patients with biliary atresia (e.g. they report that three cases with presumptive diagnosis of biliary atresia died as a result of delayed treatment).

## Examples of interventions

Only four papers in our review evaluated interventions designed to improve surgical referral and care systems in the participating countries. Three studies focused on inter-hospital coordination mechanisms [6,15,16], while the fourth one examined a surgical skills development programme [31]. A description of each study is provided in the following section.

**Improving referral and care coordination.** In Nepal, Fleming et al [15] assessed a programme implemented in a rural district hospital through a collaboration between the ministry of health and an NGO. Patients referred for surgical care at a higher-level facility were provided with financial, social and logistical support throughout the referral process, including a dedicated accompanying community health worker and full lodging for family members at the referral institution. While the programme improved patients' experience with referral care, the cost was not sustainable, leading to discontinuation of the programme. Instead, the ministry invested in building the surgical capacity of the local hospital; and the study did not include a comparative cost benefit analysis.

In Brazil, Rudge et al [16] evaluated an intervention improving system-wide coordination between a level II and a level III hospital to reduce overcrowding at the tertiary hospital and to improve safety of pregnancy. The intervention consisted of enhancing triage and diagnostic practices at the two facilities to promptly detect low-risk and high-risk obstetric patients, and

to facilitate treatment at the right level of care. This included staff training and establishment of a new cadre of specialised health workers, as well as enhancing transport and communication links between the two facilities for the efficient exchange of patients according to the complexity of the case. The intervention led to an improvement in maternal and perinatal outcomes: C-section rates decreased at the level II hospital; stabilised at the level III hospital; and perinatal mortality rates decreased in both hospitals. However, no economic analysis was carried out as part of the evaluation.

In China, Shi et al [6] examined the impact of referrals from outreach specialists on the hospitalisation costs of rural patients requiring advanced surgery at a tertiary hospital to treat digestive tract cancer. The study found that length of stay at the tertiary hospital for patients referred by the outreach specialist was lower than that of patients referred by local doctors or self-referrals. Hospitalisation and particularly diagnostic test costs were also lower than those of the other two groups. The authors suggested that outreach specialists were more qualified than local clinicians in determining the correct diagnosis and providing treatment, thereby minimising elective referrals and associated costs. However, the focus of the study was narrow, the costs of the outreach programme were not measured; and lessons learned regarding implementation (e.g. impact on service delivery, logistics, sustainability etc.) were not reported.

**Improving skills distribution in the referral system.** In Niger, Sani et al [31] evaluated an intervention to enable general physicians at rural district hospitals to provide surgical services to the local population through a targeted training programme. The study reported a large decrease in the number of emergency referrals from the intervention district hospitals to the regional hospital with the introduction of the programme in 2007. Specifically, an 82% decrease in referrals was recorded compared to the time prior to the introduction of surgical services at district level (in 2005) and a 52% decrease in referrals compared to the surgical camp model previously used by the ministry as a solution to answer the demand of surgical care at district level (in 2006). However, the costs of the training programme and cost savings from reduced referrals were not reported.

## Discussion

This scoping review of scientific evidence aimed to investigate what is already known in regards to inter-hospital surgical referral systems in LMICs. Our findings demonstrate that this field of study is a largely uncharted territory. The functionality of the surgical referral system in developing countries is, to some extent, even more critical than in developed economies as it allows for more efficient use of scarce surgical skills and resources; and continuity of care across different tiers of surgical services, which in these regions are unevenly distributed [4]. While only 14 papers met the inclusion criteria for this review, they provide a useful benchmark of the status and dearth of evidence on surgical referrals across LMICs; and on some of the commonalities. The conceptual framework developed through this review (Fig 2) aims to identify some of the critical areas that are of relevance to national decision-makers; and where further research is needed.

The first key message from the papers is that multiple shortages, especially at the first level of surgical care (the district hospital), impacts on referrals, act as obstacles to safe and appropriate referrals and as contributors to inappropriate referrals. The limitations of surgical infrastructure, equipment and personnel at first-level hospitals highlighted by the selected papers is well documented in the wider literature [4,36,37]. Evidence in our review shows that the shortage of basic essentials extends to secondary and tertiary hospitals [14], which should be better equipped considering size and remit of these facilities. This leads to a vicious cycle, where patients are referred from one hospital to another, sometimes after delays and deterioration in

patients' conditions, without guarantee that the next facility will have the resources and capacity to provide the necessary care [7], entailing risks of negative clinical outcomes. Resource shortages have historical roots, linked to the low prioritisation of surgery in public health funding in LMICs [7]. In the last five years, since the publication of the Lancet Commission on Global Surgery [4], national governments and the international community are focusing on strengthening surgical services in LMICs [31,38]. However, these efforts need to go hand in hand with interventions to improve functionality and reliability of referral pathways in order to ensure a safe and appropriate continuum of care [4].

Gaps in the surgical and decision-making skills of clinicians at sending hospitals were identified by the reviewed studies as a major driver of referrals in LMICs, including evidence of: delayed or inappropriate referrals across cadres and seniority levels, poor patient management, stabilisation and lack of basic safety measures during referrals; pointing to the need for better training, supervision and guidance for all staff involved in referral processes [14,33,34]. These findings are echoed by other studies assessing hospitals' surgical capacity, which report lack of expertise as the primary obstacle to treatment of patients at local facilities, leading to inappropriate referrals [39–41]. Emerging research in this area suggests that attitudes and motivation of surgical providers might also play a role in unnecessary referrals [41]. This is a relatively new and sensitive concept not examined by the reviewed studies, which focused on skills' levels but not professionalism, which might benefit from further research.

A second key message from the studies in our review, in line with international recommendations [4], is the need for national protocols for the triage of common urgent surgical conditions to facilitate more timely transfer of patients to appropriate levels of care [33,35]. National referral guidelines exist in some LMICs, either in general form (e.g. health-sector wide, including surgery, in India [42] and Kenya [43]) or sector-specific (e.g. maternity and neonatal care in Zambia [44]), but comprehensive protocols are still lacking in most cases. Implementation, when in place, is also an issue. As reported by den Hollander [14], protocols for referral of burn cases exist in South Africa, but pathways are not correctly followed. International protocols can be used as a template to develop national responses in countries where guidelines are lacking [33], while supporting better enforcement in countries with existing ones. Standard protocols should extend to communication practices between hospitals. Gyedu et al [32] found that structured referral forms reduced the number of missing information items, essential for the delivery of critical and emergency surgical care. In LMICs, paper-based referral forms are often the only piece of documentation following patients through the health care system. Clinician training and compliance in the completion of structured referral forms may reduce unnecessary referrals [32], as well as improving the quality of referrals. These areas require systematic research.

A third lesson from the intervention studies in this review, even if empirical evidence is limited, is that improving coordination and communication between different level facilities can be beneficial in enhancing efficiency of referral pathways. There is consensus in the literature that strengthening capacity of referring hospitals to manage more conditions locally, especially urgent conditions, could improve outcomes, decrease the need for referral and reduce the burden on tertiary facilities [11,35]. Sani et al [31] demonstrated how the introduction of a new training programme for physicians in district hospitals in Niger led to a large decrease in the number of surgical referrals. While Sani et al study is not generalisable (because results are context specific), other studies confirm the positive effect of skills development on reducing unnecessary referrals from district hospitals [45]. What needs to complement such evidence is more research that demonstrates the effectiveness and potential for comparable outcomes of delivering surgical interventions at district hospitals [46]. For those surgical cases that cannot be managed locally, it is important to improve the availability of specialists and to enhance

service capacity at higher level facilities [11,35]. As reported by Nkurunziza [35], district hospitals at times have to delay transfer of patients to higher level of care while waiting for an appointment with specialists (there was only one neurosurgeon in public hospitals in Rwanda at the time of the study). Therefore, interventions to improve referral systems should address gaps at both the sending and receiving ends of the referral pathway.

A final important message emerging from this review of 14 studies is that from a methodological perspective there is no standard approach to assessing functionality and effectiveness of referral systems in surgery. Each study focused on a different aspect of surgical care and referrals, using its own measurement tools and metrics. The World Health Organization [47] has identified functioning referral systems, including strong linkages between different levels of care and information flows, as an essential element to ensuring safe and effective services. Developing strategies for assessing referral systems and measuring their impact on access, coverage and quality of services may contribute to improving overall health systems performance.

## Limitations

A limitation of our review is the low number of rigorous studies identified and their heterogeneity, which made drawing definite conclusions difficult and highlighted the need for more empirical research. The quality of retrieved evidence was affected by risk of bias in some cases (see S2 Table), deriving from inconsistencies in data reporting (in Siraj et al [30] and Simba et al [11]); weak outcome measures (e.g. assessment of appropriateness of referrals in den Hollander et al [14] and Goodman et al [13]); only one investigator involved in the data collection (in Crandon et al [33] study); and failure to account for confounding factors (in Nkurunziza et al [35] study some of the patients might have benefitted from an NGO- support programme available at the time of the study in the participating hospitals).

## Supporting information

**S1 Table. PRISMA checklist for scoping reviews.**
(DOCX)

**S2 Table. Search string.**
(DOCX)

**S3 Table. Characteristics of studies included in the review.**
(DOCX)

## Acknowledgments

We are particularly grateful to Paul Murphy, Library of the Royal College of Surgeons in Ireland, for providing technical advice on the early development of the search strategy for this review, and to Prof Chris Lavy, Nuffield Department of Orthopaedics, Rheumatology and Musculoskeletal Sciences, University of Oxford, for reviewing the draft manuscript and offering valuable expert opinion and insight.

## Author Contributions

**Conceptualization:** Chiara Pittalis, Ruairi Brugha, Jakub Gajewski.

**Data curation:** Chiara Pittalis.

**Formal analysis:** Chiara Pittalis, Ruairi Brugha, Jakub Gajewski.

**Investigation:** Chiara Pittalis, Ruairi Brugha, Jakub Gajewski.

**Methodology:** Chiara Pittalis, Jakub Gajewski.

**Project administration:** Chiara Pittalis.

**Software:** Chiara Pittalis.

**Visualization:** Chiara Pittalis.

**Writing – original draft:** Chiara Pittalis, Ruairi Brugha, Jakub Gajewski.

**Writing – review & editing:** Chiara Pittalis, Ruairi Brugha, Jakub Gajewski.

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
