## [Decision Letter · Decision Letter 0]

21 Aug 2019

PONE-D-19-17583

Surgical referral systems in low- and middle- income countries: a review of the evidence

PLOS ONE

Dear Ms Pittalis,

Thank you for submitting your manuscript to PLOS ONE. After careful consideration, we feel that it has merit and would like to accept it for publication after you have addressed the minor revision comments raised by the reviewers.   Therefore, we invite you to submit a revised version of the manuscript that addresses the points raised during the review process.

We would appreciate receiving your revised manuscript by Oct 05 2019 11:59PM. To enhance the reproducibility of your results, we recommend that if applicable you deposit your laboratory protocols in protocols.io, where a protocol can be assigned its own identifier (DOI) such that it can be cited independently in the future. For instructions see: http://journals.plos.org/plosone/s/submission-guidelines#loc-laboratory-protocols

We look forward to receiving your revised manuscript.

Kind regards,

Irene Agyepong

Academic Editor

PLOS ONE

Journal Requirements:

1. Please state in your methods section the start and end date of searches.

Reviewers' comments:

Reviewer's Responses to Questions

**Comments to the Author**

1. Is the manuscript technically sound, and do the data support the conclusions?

Reviewer #1: Yes

Reviewer #2: Yes

2. Has the statistical analysis been performed appropriately and rigorously? 

Reviewer #1: N/A

Reviewer #2: N/A

3. Have the authors made all data underlying the findings in their manuscript fully available?

Reviewer #1: Yes

Reviewer #2: Yes

4. Is the manuscript presented in an intelligible fashion and written in standard English?

Reviewer #1: Yes

Reviewer #2: Yes

5. Review Comments to the Author

Reviewer #1: The manuscript investigates current knowledge regarding inter-hospital surgical referral systems in LMIC. Some of the papers identified and selected from the search do not provide detailed information to answer the research questions. Nevertheless it is an important effort in highlighting the need for further research in the area of inter-hospital surgical referral in LMIC. Table S3 highlights the risk of bias and limitations of the included studies.

Reviewer #2: This is a well written manuscript that sought to investigate current knowledge regarding inter-hospital surgical referral systems in LMICs; to identify the main obstacles to their functioning and to critically assess proposed solutions. The authors used a scoping review approach and applied relevant guidelines and tools such as the Cochrane guidelines, Joanna Briggs Institute guidelines, the Preferred Reporting Items for Systematic Reviews in their review. The results are discussed appropriately under health facility factors, health system factors and environmental factors. The authors identified interventions that can improve surgical referrals such as the need to improve referral and coordination of care and improving skills distribution in the referral system. The authors discuss their results in the light of the present body of knowledge and conclude that among others, improving coordination and communication between different level facilities can improve surgical referral systems. They further advocate for a standard methodological approach to assessing functionality and effectiveness of referral systems in surgery.

This manuscript can be improved by addressing the following comments:

1. Since tables must be stand alone and understandable, please explain what JBI means in the table “S2 Table. Characteristics of studies included in the review”.

2. Page 20, line 282, please check for typographical errors.

3. Page 29, line 296, please check for typographical errors.

4. How is reference 28 as mentioned in text used in the evaluation of the quality of evidence? The table S2 sugggests that JBI which is reference 27 was used. Please clarify this.

5. How many and which papers are you referring to in this statement “some of them mention that surgery is not typically among the top priorities for local governments, with implications…… (page 20 line 280 -281)?’’

6. How many and which papers are you referring to in this statement ‘A number of studies pointed to the influence of wider country contexts……?’. Are the examples you cite in the text that follows all the papers referred to in this statement or are there more?

7. For the table ‘S1 Table: Search string’, present the search string for the other databases besides MEDLINE which were also searched in order for readers to know and understand how the search stratergy was adopted to these databases to facilitate reproducibility of your search.

6. PLOS authors have the option to publish the peer review history of their article (what does this mean?). If published, this will include your full peer review and any attached files.

Reviewer #1: Yes: Edward Antwi

Reviewer #2: Yes: Hannah Brown Amoakoh

---

## [Author Response · Author response to Decision Letter 0]

9 Sep 2019

09 September 2019

Dear Editor,

we would like to take this opportunity to thank you and the Reviewers for the positive feedback on our manuscript and for the valuable comments and suggestions for improvement. As requested, please find below our responses and clarifications.

Best wishes on behalf of the authors,

Chiara Pittalis

Comments:

1. Please state in your methods section the start and end date of searches.

R: thank you for the advice, we have added the dates to the manuscript.

1. Since tables must be stand alone and understandable, please explain what JBI means in the table “S2 Table. Characteristics of studies included in the review”.

R: we have modified the table heading accordingly.

2. Page 20, line 282, please check for typographical errors.

R: thank you for spotting the typo, we have corrected it.

3. Page 29, line 296, please check for typographical errors.

R: we have corrected the typo accordingly.

4. How is reference 28 as mentioned in text used in the evaluation of the quality of evidence? The table S2 suggests that JBI which is reference 27 was used. Please clarify this.

R: reference no. 27 is the Thaddeus and Maine ‘three delays’ model, which informed the thematic analysis of the papers as described in the ‘Data charting and analysis’ section. We are assuming the confusion may be between the JBI tools and reference no. 28 – Atkins et al paper on how to grade quality of evidence and strength of recommendations. We used the JBI tools and methods to conduct an in-depth analysis of the reviewed papers, the outcomes of which are presented in the Supplementary Table. 

The Atkins et al framework categorises type of evidence as high, low and very low, and it helped us summarise in a simple and systematic way our overall reflections on the strength of the studies being assessed and confidence placed in their recommendations. This is mentioned in lines 220-221, pg. 12 of our manuscript. Since all reviewed studies were observational and mostly descriptive in nature, the quality of evidence/recommendations was graded as low.

5. How many and which papers are you referring to in this statement “some of them mention that surgery is not typically among the top priorities for local governments, with implications…… (page 20 line 280 -281)?’’

R: in our review this statement derives from Albutt et al paper (we have modified lines 280-281 in the manuscript to clarify), but the issue of lack of financing for surgery in Africa is supported by the wider literature. One good report is the paper by Citron, Chokotho and Lavy published in 2016: ‘Prioritisation of surgery in the national health strategic plans of Africa: a systematic review’ (DOI:https://doi.org/10.1016/S0140-6736(15)60848-0).

6. How many and which papers are you referring to in this statement ‘A number of studies pointed to the influence of wider country contexts……?’. Are the examples you cite in the text that follows all the papers referred to in this statement or are there more?

R: yes, the six examples in our manuscript are the ones we referred to in that opening statement (line 290 onwards). 

7. For the table ‘S1 Table: Search string’, present the search string for the other databases besides MEDLINE which were also searched in order for readers to know and understand how the search strategy was adopted to these databases to facilitate reproducibility of your search.

R: as requested, we have added all search strings to the Supplementary Table.

---

## [Editor Report · Decision Letter 1]

19 Sep 2019

Surgical referral systems in low- and middle- income countries: a review of the evidence

PONE-D-19-17583R1

Dear Dr. Pittalis,

We are pleased to inform you that your manuscript has been judged scientifically suitable for publication and will be formally accepted for publication once it complies with all outstanding technical requirements.

With kind regards,

Irene Agyepong

Academic Editor

PLOS ONE
---

## [Editor Report · Acceptance letter]

20 Sep 2019

PONE-D-19-17583R1 

Surgical referral systems in low- and middle-income countries: a review of the evidence 

Dear Dr. Pittalis:

I am pleased to inform you that your manuscript has been deemed suitable for publication in PLOS ONE. Congratulations! Your manuscript is now with our production department. 

With kind regards,

on behalf of

Dr. Irene Agyepong 

Academic Editor

PLOS ONE